# Subtypes of the Completely Reversed Flow Waveform in Vertebral Artery Can Help to Differentiate Subclavian Artery Occlusion from Severe Stenosis

**DOI:** 10.3390/diagnostics13010146

**Published:** 2023-01-01

**Authors:** Shun-Ping Chen, Zhen Zhong, Dong-Pei Tu

**Affiliations:** Department of Ultrasonography, The First Affiliated Hospital of Wenzhou Medical University, Wenzhou 325000, China

**Keywords:** subclavian artery steal, subclavian artery stenosis, vertebral artery, completely reversed flow, ultrasound, Doppler

## Abstract

Objectives: To investigate the value of subtypes of completely reversed flow (CRF) waveform in vertebral artery (VA) on Doppler ultrasound in differentiation occlusion from severe stenosis of the ipsilateral proximal subclavian artery (SA). Methods: A total of 357 patients with CRF in the VA on Doppler US were reviewed retrospectively. Among them, 49 patients (mean age, 68.2 ± 7.8 years) confirmed by digital subtraction angiography (DSA) were included. According to the status of diastolic flow, the CRF was divided into continuous CRF (CCRF, *n* = 27) and intermittent CRF (ICRF, *n* = 22). The correlation of subtypes of CRF waveform and VA parameters with the severity of SA stenosis was evaluated. The severity of SA stenosis was determined by DSA. Results: Of those 49 patients, SA occlusion was observed in 33 patients (67%, occlusion group) and severe stenosis in 16 patients (33%, stenosis group). The subtypes of CRF waveforms showed a significant between-group difference (*p* = 0.005). CCRF exhibited an accuracy of 85.2% (23/27) in diagnosing SA occlusion. The diameter of the target VA with ICRF showed a significant between-group difference (*p* = 0.041). The target VA diameter ≥ 3.8 mm in ICRF achieved an accuracy of 81.8% (18/22), and its combination with CCRF achieved an accuracy of 83.7% (41/49) in the differentiation of SA occlusion from severe stenosis. Conclusions: Subtypes of CRF in VA can help to differentiate SA occlusion from severe stenosis. CCRF has higher accuracy in diagnosing SA occlusion. The CCRF waveform plus VA diameter in ICRF is more accurate for differentiating SA occlusion from severe stenosis.

## 1. Introduction

Vertebrobasilar (posterior) circulation ischemia (PCI) occurs in up to 20% of patients with stroke [1]. The mortality caused by PCI-involved stroke is as high as 21% [2]. PCI often arises from severe subclavian artery (SA) stenosis or occlusion. Percutaneous transluminal angioplasty (PTA), with or without stenting, achieves a success rate of 100% in treating symptomatic patients with SA stenosis [3,4,5] but just 48% to 87% for patients with complete SA occlusion [3,4,5,6,7]. Those data suggest that severe SA stenosis may just be treated with PTA, whereas some total occlusion may fail with PTA or require surgical treatment. Thus, differentiating occlusion from severe stenosis of SA is important before treatment, as it may guide the clinician to further make better treatment strategies.

Digital subtraction angiography (DSA) is the gold standard for evaluating stenosis or occlusion of the subclavian artery. However, Doppler ultrasound is the most commonly used imaging modality for evaluating SA stenosis, either directly (by evaluating the SA) or indirectly (by evaluating the vertebral artery) [8], due to its cost-effective and noninvasive features. Doppler ultrasound has been frequently used in the intertransverse segment of the vertebral artery (VA) to obtain spectral Doppler waveform of the VA and determine if the blood flow is antegrade or retrograde in the VA [9,10,11,12,13]. The waveforms of VA on Doppler ultrasound were classified into five types, each correlating with a certain degree of SA stenosis [14,15]. Among those types of waveforms, the completely reversed flow (CRF, also called completely subclavian steal phenomenon, SSP) in the VA may indicate ipsilateral to proximal occlusion or severe stenosis of the subclavian or innominate artery before the VA origin [14,15]. In addition, CRF is further subdivided into continuous CRF (CCRF) and intermittent CRF (ICRF) in our previous study [15]. However, the value of subtypes of CRF in VA in the differentiation SA occlusion from severe stenosis is uncertain. We hypothesize that subtypes of CRF waveform in VA are helpful for the differentiation of SA occlusion from severe stenosis. Thus, the present study aimed to investigate the value of the subtypes of CRF waveforms (CCRF or ICRF) of the VA and VA parameters in the differentiation SA occlusion from severe stenosis. DSA was performed to determine the severity of SA stenosis.

## 2. Methods

### 2.1. Patients

This retrospective study was approved by the Review of Ethics Committee in Clinical Research of the First Affiliated Hospital of Wenzhou Medical University (Approval Date: 2 April 2022; Approval Number: KY2022-R046). All US and DSA examinations, including techniques performed for VA and SA evaluation, were performed for clinical purposes unrelated to this investigation. The requirement for informed patient consent for this retrospective review was waived.

From January 2004 to December 2021, a total of 348,976 patients were examined with VA US in our hospital. Among them, 357 patients with CRF on VA US were reviewed retrospectively. The flowchart for patient selection is shown in Figure 1. This study included 49 patients (39 men and ten women; mean age, 68.2 ± 7.8 years; range, 54–86 years). Of those 49 patients, some data of sixteen patients with CRF in VA had already been published in our previous study [15]. However, in that study, the CRFs in the VA of those 16 patients were only divided into continuous and intermittent CRF. The clinical value of subtypes of CRF in the 16 patients had not been investigated because of its smaller study cohort. Thus, another 33 patients with CRF in VA were added. The CRFs were also divided into continuous and intermittent CRF in the present study. Based on this larger cohort (49 patients), we investigated the clinical value of subtypes of CRF. The inclusion criteria were: (1) the patient had a CRF waveform in the mid-portion of the VA extracranial segment on Doppler US; (2) the patient had received DSA of the conventional aortic arch and/or selective SA; (3) the interval between Doppler examination and DSA was no more than two months; (4) the patients did not have severe heart disease with a low ejection fraction that would have resulted in reduced blood flow through the VA; (5) atherosclerosis was the cause of proximal one-sided SA stenosis or occlusion. Demographic data, clinical histories and outcomes were obtained from the hospital’s electronic medical information database.

### 2.2. Doppler Sonography

Doppler sonography was performed by experienced vascular radiologists. The patient was placed in the supine position using standard sonographic equipment (HDI 5000 and IU 22; Philips Healthcare, Bothell, WA 98021, USA; Acuson Sequoia 512; Siemens Medical Solutions, Mountain View, CA 94043, USA; GE Logiq E9, GE Healthcare, Wauwatosa, WI, USA; Hitachi Hi Vision Avius, Hitachi Aloka Medical, Tokyo, Japan) with a 4.0–9.0 MHz linear probe. The VA waveform was captured in the mid-portion of the VA extracranial segment from an angle of 60° or less [16].

CRF was defined as an utterly retrograde flow in each cardiac cycle [17]. Based on whether the end-diastolic flow velocity of the CRF was greater than 0 cm/s, the CRF cases were subdivided into two types: CCRF (*n* = 27; Figure 2A) and ICRF (*n* = 22; Figure 2B). Parameters of the reversed flow in the target VA included peak reversed velocity (PRV), end-diastolic velocity (EDV), resistant index (RI, RI = (PRV-EDV)/PRV) and diameter. The antegrade flow parameters in the VA on the contralateral side included: peak systolic velocity (PSV), EDV, RI and diameter. The measurements of at least three consecutive heartbeats were acquired. The interval between US and DSA examinations was 15 days (interquartile range (IQR), 8–34 days; range, 0–56 days).

### 2.3. Angiography

Angiography was performed on the conventional aortic arch or selective SA using intra-arterial digital subtraction techniques. A 5-F catheter was inserted into the femoral artery through the Seldinger approach. The patient’s previous DSA records were used as a reference. SSP was defined as reversed blood flow in the VA ipsilateral artery to proximal high-grade stenosis or occlusion of the subclavian or innominate artery before the VA origin. The degree of SA stenosis in the VA was rated according to the lumen diameter at the point of maximum stenosis at a disease-free segment of the subclavian artery distal to the stenosis. Complete occlusion was defined when the contrast was not observed in the lumen of the artery.

### 2.4. Statistical Analysis

Statistical analysis was performed using the SPSS 18.0 software package (SPSS Inc, Chicago, IL, 60606, USA). Categorial variables were presented as numbers and (or) percentages and compared using the chi-square test and Fisher’s exact test. Normally distributed data were presented as mean ± standard deviation (SD) and compared using a two-sample *t*-test. Non-normally distributed data were presented as median (IQR) and compared using the Mann–Whitney *U* test. The results were considered significant for *p* < 0.05. Receiver operating characteristic (ROC) curves of significant parameters in the VA were constructed to obtain the best cutoff value for differentiation of SA occlusion from severe stenosis. The best cutoff was correlated with the highest accuracy estimated by the maximal Youden index.

## 3. Results

### 3.1. Clinical Characteristics of Patients with CRF and SA Stenosis

The clinical characteristics of patients are shown in Table 1. Among those 49 patients, 39 exhibited symptoms (dizziness in 21 cases, upper limb fatigue in 9 cases, blurred vision in 1 case and other symptoms (such as loss of consciousness) in 8 cases), and ten patients were without symptoms. The interval between the US examination and the patients’ complaint was 30 (7–365) days (three days to ten years). Among the 49 patients, 22 had been treated with SA stent implantation (20 successes and two failures), two with PTA, 24 with medication, and one without treatment and follow-up.

According to DSA results, the degree of SA stenosis was classified into <70% (*n* = 0), ≥70% (*n* = 16; range, 70%−99%, 86.5% ± 8%), and 100% (*n* = 33). Thus, only SA severe stenosis and occlusion were observed in CRF patients. Using CRF for diagnosing SA occlusion or severe stenosis, the PPV values were 67.3% (33/49) and 36.5% (16/49), respectively. The interval between US and DSA was 13 (7–34) days in the severe stenosis group and 16 (9–24) days in the occlusion group (*p* = 0.161).

In total, 13 patients were accompanied by carotid or (and) severe VA stenosis or (and) occlusion confirmed by DSA. There was no significant difference in the incidence between the two groups (Table 2, *p* > 0.05).

### 3.2. Correlation of CRF Waveform and VA Parameters with Ipsilateral SA Occlusion or Severe Stenosis

The CRF types and VA parameters were compared between the two groups (Table 3). There were no significant differences in age, gender and VA parameters (*p* > 0.05). However, the waveform type of CRF was correlated with the SA occlusion and severe stenosis (*p* = 0.005, Figure 3). In using the CCRF waveform to differentiate SA occlusion from severe stenosis, the sensitivity, specificity, positive predictive value (PPV), negative predictive value (NPV), and accuracy were 69.70% (23/33), 75.00% (12/16), 85.19% (23/27), 54.55% (12/22) and 71.43% (35/49), respectively. There were four false PPVs and ten false NPVs.

### 3.3. Diagnostic Values of VA US Parameters in 27 Patients with VA CCRF

Table 4 shows no significant differences in age, gender, site, VA diameter, PSV and RI in VA between the SA occlusion and severe stenosis groups. However, CCRF achieved an accuracy of 85.2 % (23/27) in the differentiation of SA occlusion from severe stenosis.

### 3.4. Diagnostic Values of VA US Parameters in 22 Patients with VA ICRF

Table 5 shows no significant differences in age, gender, site, PSV, VA RI and contralateral VA diameter between the SA occlusion and severe stenosis groups. However, the target VA diameter significantly differed between the two groups (*p* = 0.041). Using a target VA diameter of 3.8 mm as a cutoff value for differentiation SA occlusion from severe stenosis, the sensitivity, specificity, PPV, NPV and accuracy were 80% (8/10), 83.3% (10/12), 80% (8/10), 83.3% (10/12) and 81.8% (18/22), respectively.

### 3.5. Diagnostic Value of CCRF Combined with the Diameter of the Target VA with ICRF

When combined with the CCRF waveform and diameter in the target VA of ICRF for differentiation SA occlusion from severe stenosis, the accuracy was 83.7% (41/49), which was higher than those of the CRF waveform (using CRF as a positive predictor) and the CCRF waveform (using CCRF as a positive predictor) for the differentiation of SA occlusion from severe stenosis (83.7% (41/49) vs. 67.3% (33/49) vs. 71 % (35/49)). However, no significant differences were observed among the three methods for differentiation SA occlusion from severe stenosis (*p* > 0.05).

## 4. Discussion

Our results showed that in VA CRF patients, the incidence of ipsilateral SA occlusion (67%, 33/49) was higher than that of server SA stenosis (33%, 16/49), which is not consistent with a previous report that CRF in ultrasound is 100% correlated with SA occlusion (3/3 or 19/19), except for VA occlusion [13,18]. Sakima et al. [14]. have reported that both SA middle-grade stenosis and occlusion cause CRF in five CRF cases. However, our previous and present studies confirm that VA CRF is only associated with severe SA stenosis and occlusion [15]. Considering the larger sample size in this study, we think our results are more reliable than those reported before [13,14,18].

In addition, the current study showed that CCRF could diagnose SA occlusion with higher accuracy (85.2%, 23/27) but another ICRF with a lower accuracy (45%, 10/22). To the best of our knowledge, subtypes of CRF waveform have never been used to differentiate SA occlusion from severe stenosis. The pressure differs between proximal (before the origin of VA) and distal SA (after the origin of VA). As the proximal SA is occluded, the pressure of distal SA is not affected by the proximal SA, which can not disturb the reversed blood flow in the target VA and appear as CCRF in the VA. In severe SA stenosis, a gradient pressure emerges between the proximal and distal SA, disturbing the reversed blood flow in the target VA; in this condition, ICRF appears on Doppler US. However, this hypothesis needs further confirmation.

Next, we also determined that the target VA diameter greater than or equal to 3.8 mm achieved a high accuracy of 81.8% (8/22) in the differentiation of SA occlusion from severe stenosis in patients with ICRF. A study [18] has reported that the vertebral lumen is wider in cases with SA occlusion than in cases with SA stenosis; however, in this study, bidirectional flow in VA was also included for analysis, and the CRF was not divided into CCRF and ICRF. As the SA occlusion, we suppose a more reversed flow in the VA is needed to support the blood into the affected upper limbs, thus increasing the diameter and velocity of the target VA. Additionally, this hypothesis needs further validation.

The velocity in the SA may be used to evaluate the degree of SA stenosis [19,20]. However, the SA, especially the left SA, is difficult to be detected by US. Meanwhile, almost all VA in the intertransverse segment could be detected by US [10,11]. Thus, typical features of subclavian steal on US, such as CRF, can be relied on to diagnose proximal SA stenosis or occlusion. The subtype of the CRF provides higher accuracy for the differentiation of SA occlusion from severe stenosis. Since CCRF and ICRF are two types of waveforms, their diagnosis performance cannot be affected by heart diseases, such as heart failure or arrhythmia. Last, both parameters can be used in the bedside US examination for evaluating SA stenosis in an emergency. Additionally, both predictive subtypes are particularly useful in patients who cannot endure DSA examination or are allergic to contrast agents.

This study has some limitations. First, this retrospective study could have introduced a selection bias. Further prospective studies with larger numbers of patients are needed to determine the diagnostic efficiencies of CCRF, ICRF and VA diameter. Second, we did not consider the effect of VA hypoplasia on SA stenosis, though it may alleviate SA stenosis, as previously reported [21]. Third, because of a smaller number of patients with right SA (only five cases), the CRFs type in the VA of those 49 patients were not subdivided into the left and right SA for the between-group analysis. The diagnostic value of CCRF for SA at different sides should be investigated in the future. Fourth, DSA confirmed all patients with CRF in VA, and most patients had circulatory symptoms; thus, the diagnostic values of the parameters in symptomless patients need further study. Fifth, all of the patients included in this study had atherosclerosis-related stenosis or occlusion of the SA; therefore, the diagnostic values of the parameters for SA stenosis with other etiologies (e.g., dissection and Takayasu arteritis) remain to be explored.

## 5. Conclusions

On Doppler ultrasound, subtypes of CRF in VA can help differentiate SA occlusion from severe stenosis. CCRF has a higher accuracy in diagnosing SA occlusion. The target VA diameter ≥ 3.8 mm in ICRF achieved higher accuracy for differentiation SA occlusion from severe stenosis. The CCRF waveform plus VA diameter in ICRF is more accurate for the differentiation of SA occlusion from severe stenosis. CRF subtypes can be used in bedside US examinations, an emergency or in patients who cannot endure a DSA examination or are allergic to contrast agents, which may inform the clinician about the degree of SA stenosis and aid clinical decision making. Future prospective studies with a larger number of patients are needed to determine further the effectiveness of CRF subtypes combined with VA parameters in the differentiation of SA occlusion from severe stenosis.

## Figures and Tables

**Figure 1 diagnostics-13-00146-f001:**
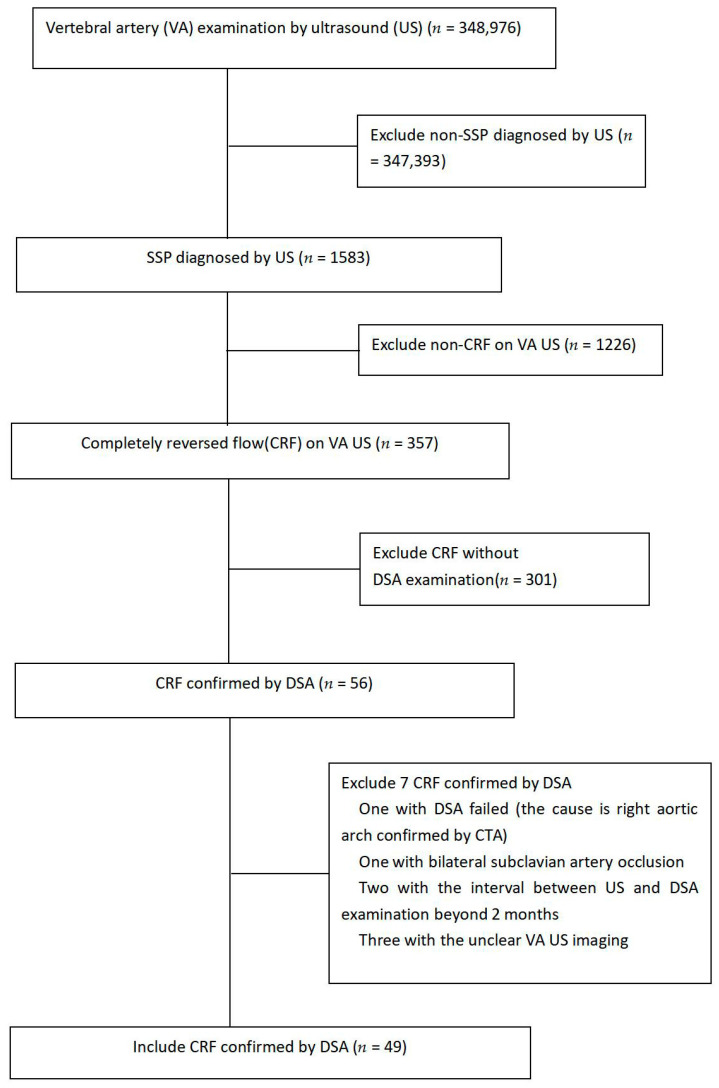
Flowchart of criteria for patient selection. VA, vertebral artery; US, ultrasound; SSP, subclavian steal phenomenon; CRF, completely reversed flow; DSA, digital subtraction angiography; CTA, computer tomography angiography.

**Figure 2 diagnostics-13-00146-f002:**
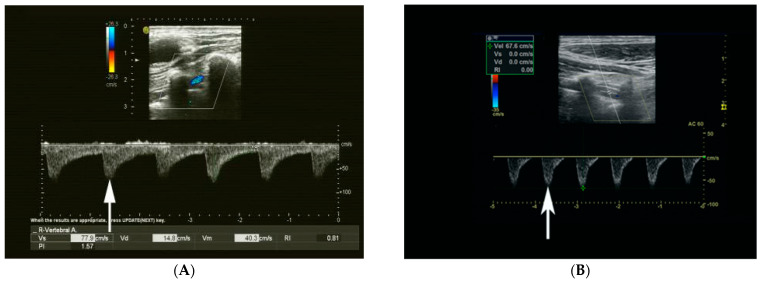
(**A**) A 60−year−old man with CCRF (arrow) in the right VA on Doppler US imaging. DSA confirmed occlusion in the ipsilateral proximal SA. CCRF, continuous completely reversed flow; VA, vertebral artery; US, ultrasound; DSA, digital subtraction angiography; SA, subclavian artery. (**B**) A 61−year−old man with ICRF (arrow) in the left VA on Doppler US imaging. DSA confirmed severe stenosis in the ipsilateral proximal SA (95%). ICRF, intermittent completely reversed flow; VA, vertebral artery; US, ultrasound; DSA, digital subtraction angiography; SA, subclavian artery.

**Figure 3 diagnostics-13-00146-f003:**
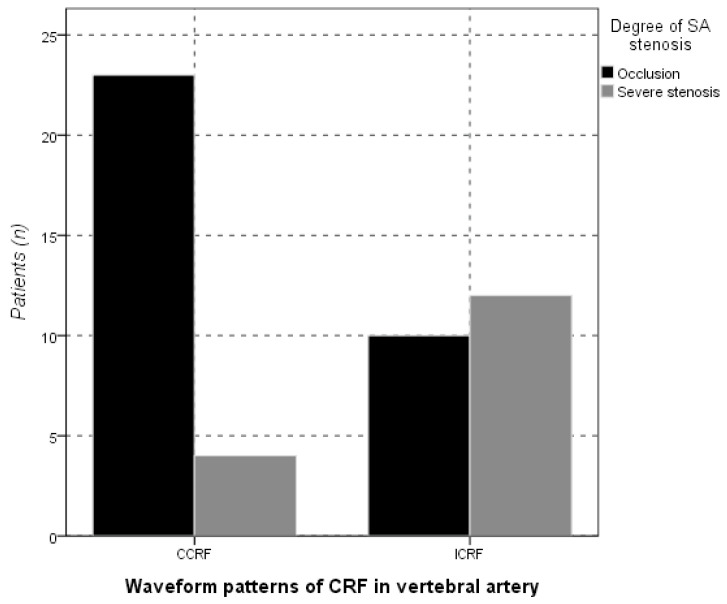
Comparison of subtypes of the completely reversed flow (CRF) waveform in vertebral artery (VA) and severe stenosis or occlusion of the subclavian artery (SA). SA occlusion was associated with continuous CRF (CCRF), and intermit CRF (ICRF) was involved with severe SA stenosis (*p* = 0.005). CRF, completely reversed flow; VA, vertebral artery; SA, subclavian artery; CCRF, continuous CRF; ICRF, intermittent CRF.

**Table 1 diagnostics-13-00146-t001:** Clinical characteristics of 49 patients with CRF in the VA.

Characteristics	Datum
Clinical characteristics	
Age (y)	54–86 (68.2 ± 7.8)
Gender (male/female)	39/10
Hypertension	27 (55%)
Hyperlipidemia	7 (14.3%)
Diabetes	15 (30.6(%)
Smoking habit	11 (22.4%)
Obesity	15 (30.6%)
Concomitant diseases	
Coronary artery diseases	3 (6.1%)
Previous myocardial infarction	1 (2%)
Previous stroke/TIA	7 (14.3%)
Clinical symptoms and physical examination	
Asymptomatic patients	10 (20.4%)
Symptomatic patients	39 (79.6%)
Symptoms of posterior fossa ischemia	22 (44.9%)
Upper limb claudication	9 (18.4%)
Pulse weakness	33 (67.3%)
Mean SBP difference (mmHg) *	33 (16–46)
Mean DBP difference (mmHg) *	9 (5.5–18)

CRF, completely reversed flow; VA, vertebral artery; TIA, transient Ischemic attack; SBP, systolic blood pressure; DBP, diastolic blood pressure.* Only 21 patients with difference in mean SBP and DBP were recorded.

**Table 2 diagnostics-13-00146-t002:** Concurrent carotid or (and) VA severe stenosis or (and) occlusion in 49 patients of two subtypes of CRF.

Events	CCRF Group (*n* = 27)	ICRF Group (*n* = 22)	*p*-Value
Carotid artery occlusion	3	3	0.805
Vertebral artery occlusion	2	3	0.497
Carotid and vertebral artery occlusion	1	1	0.885

VA, vertebral artery; CRF, completely reversed flow; CCRF, continuous CRF; ICRF, intermittent CRF.

**Table 3 diagnostics-13-00146-t003:** Comparison of two subtypes of CRF and VA parameters between the SA occlusion group and the severe stenosis group.

Parameters	Occlusion (*n* = 33)	Severe Stenosis (*n* = 16)	*p*-Value
Age (years)	67.7 ± 7.8	69.2 ± 8.0	0.585
Gender (male/female)	25/8	14/2	0.464
Site (left/right)	28/5	16/0	0.158
Waveform type			0.005
CCRF	23	4	
ICRF	10	12	
Diameter (mm)			
TVA	3.4 ± 0.7	3. 3 ± 0.7	0.807
CVA	3.8 ± 0.8	3.7 ± 0.6	0.411
PSV (cm/s)			
TVA (PRV)	63 (41–84)	50 (40–77.25)	0.337
CVA	76 (60.5–101)	79.5 (61–91.25)	0.725
Resistant index (RI)			
TVA	0.88 ± 0.11	0.96 ± 0.07	0.051
CVA	0.73 ± 0.10	0.74 ± 0.12	0.492

CRF, completely reversed flow; VA, vertebral artery; SA, subclavian artery; CCRF, continuous CRF; ICRF, intermittent CRF; TVA, target VA; CVA, contralateral VA; PSV, peak systolic velocity; PRV, peak reversed velocity; RI, resistance index.

**Table 4 diagnostics-13-00146-t004:** Parameters of VA ultrasound in 27 patients with CCRF in the VA between the SA occlusion group and the severe stenosis group.

Parameters	Occlusion Group (*n* = 23)	Severe Stenosis Group (*n* = 4)	*p*-Value
Age (years)	67.0 ± 7.2	71.3 ± 6.7	0.284
Gender (male/female)	18/5	3/1	1.000
Site (left/right)	18/5	4/0	0.561
Diameter (mm)			
TVA	3.2 ± 0.7	3. 8 ± 0.5	0.096
CVA	3.8 ± 0.8	3.9 ± 0.6	0.825
PSV (cm/s)			
TVA (PRV)	60 (40–84)	76 (47.25–84)	0.473
CVA	76 (59–99)	84 (68.75–97.25)	0.682
Resistant index (RI)			
TVA	0.83 ± 0.09	0.84 ± 0.04	0.800
CVA	0.71 ± 0.09	0.75 ± 0.09	0.441

VA, vertebral artery; CCRF, continuous completely reversed flow; SA, subclavian artery; TVA, target VA; CVA, contralateral VA; PSV, peak systolic velocity; PRV, peak reversed velocity; RI, resistance index.

**Table 5 diagnostics-13-00146-t005:** Parameters of VA ultrasound in 22 patients with ICRF in the VA between the SA occlusion group and the severe stenosis group.

Parameters	Occlusion (*n* = 10)	Severe Stenosis (*n* = 12)	*p*-Value
Age (years)	69.3 ± 9.1	68.5 ± 8.5	0.833
Gender (male/female)	7/3	10/2	0.624
Diameter (mm)			
TVA	3.8 ± 0.7	3. 2 ± 0.7	0.041
CVA	3.7 ± 0.9	3.6 ± 0.6	0.849
PSV (cm/s)			
TVA (PRV)	66 (47.25–98.25)	46 (40–65.5)	0.099
CVA	75.5 (67.75–114)	78 (58.75–90.5)	0.664
Resistant index (RI)			
CVA	0.79 ± 0.12	0.74 ± 0.13	0.363

VA, vertebral artery; ICRF, intermittent completely reversed flow; SA, subclavian artery; TVA, target VA; CVA, contralateral VA; PSV, peak systolic velocity; PRV, peak reversed velocity; RI, resistance index.

## Data Availability

The data supporting this study’s findings are available on request from the corresponding author. The data are not publicly available due to privacy or ethical restrictions.

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
