# Peer review of "Subtypes of the Completely Reversed Flow Waveform in Vertebral Artery Can Help to Differentiate Subclavian Artery Occlusion from Severe Stenosis"

_diagnostics, 2023, doi:10.3390/diagnostics13010146_

Round 1

Reviewer 1 Report

Current study evaluated the prevalence of complete subclavian artery occlusion in two differential types of CCRF and ICRF. The presence of CCRF was significantly associated with the presence of total occlusion of ipsilateral subclavian artery. In addition, target VA diameter also had a diagnostic value in the differenciation of SA occlusion from severe stenosis. I would like to congratulate authors on the successful research achievements.

While I agree with the overall findings, I would like to add a few minor points.

1) The abbreviation was used in the figure 1, 3 (and legends), but there is no explanation for the abbreviation.

2) Usually, subclavian steal phenomenon or subclavian artery stenosis is more common in left side. Although, authors showed the information in Table 3 & 4, the diagnostic accuracy of CCRF is different in the presence of left/right subclavian artery occlusion ?

3) Current study is evaluated from the patients who had subclavian steal phenomenon in ultrasound VA examination. Therefore, some patients with subclavian occlusion or severe stenosis without SSP might be not included in current study. Is there any patients who had confirmed subclavian artery occlusion with DSA, but they did not show SSP ? 

Reviewer 2 Report

Dear authors

Thank you for submitting your article in journal. Your research article is very interesting because the investigation to diagnosis of SA occlusive disease in PCI before revascularization is very important. However, some patients had the limitation to performed the DSA or CTA. A lot of question still debate and controversies about mainstay of treatment and choice for revascularization in SA occlusive disease. I have some comments on your manuscript.

Major comment

1.     Introduction > statement “As a result, severe SA stenosis is usually treated using PTA, whereas 32 total occlusion may require surgical treatment.” [line 32-33]

-      Currently, not only the severe stenotic lesion of SA but also occlusion of SA may prefer to be treated with endovascular therapy first due to risk of open surgery of truncal arteries. So, this sentence may need the cited to the references or up to date. In addition, the differentiation occlusion from severe stenosis of SA by the Doppler ultrasound may not crucial as it affects treatment methods because the lesion characteristic, length, calcification which determine the choice of treatment (open surgery versus endovascular therapy) cannot full evaluate by Doppler ultrasound. (Especially when limited by body habitus)

-      We suggest the author to revise this statement which may cause of debate between the method to choose the proper modalities of treatment of SA occlusive disease.

2.     Introduction > statement “That is, differentiation occlusion from severe stenosis of SA is crucial as it affects treatment methods. Therefore, efficient modalities should be explored to enable accurate and noninvasive diagnosis of severe SA stenosis or complete SA occlusion.” [line 33-36]

-      As the author use the duplex ultrasound to evaluate SA stenosis/occlusion So, the disadvantage of gold standard tools of the disease by Digital subtraction angiography (DSA) should be described in the introduction part to encourage the readers and physician that “Why the Duplex ultrasound to differentiate the lesion (occlusion/stenosis) of SA are important in clinical practice ?”

3.     Introduction > statement “As a result, severe SA stenosis is usually treated using PTA, whereas 32 total occlusion may require surgical treatment.” [line 32-33]

-      Currently, not only the severe stenotic lesion of SA but also occlusion of SA may prefer to be treated with endovascular therapy first due to risk of open surgery of truncal arteries. So, this sentence may need the cited to the references or up to date.

4.     Conclusion > The author should describe the implication to practice. 

                                               i.     Could the accuracy of the Doppler ultrasound to differentiate the type of SA lesion replace the DSA to diagnose the SA lesion?

                                             ii.     How the result of this diagnostic tool to differentiate the type of lesion of SA can change the clinical practice? 

5.     Conclusion > statement “CRF subtypes can be used in the bedside US examination, the emergency, or in patients who cannot endure DSA examination or are allergic to contrast agents.” [line 263-265]

-      The author should describe about (1) “Is it safe for replacing of DSA by Doppler ultrasound to decide the further management in patients with SA occlusive disease induced PCI.” (2) The ultrasound guide endovascular therapy or bypass surgery of SA lesion without DSA may increase risk of perioperative  complications.  

Minor comments 

1.     All abbreviations in the main manuscript (not included the abstract) should be written in full text before the abbreviation. Example: CRF

2.     All abbreviations (CRF, SSP, DSA, CTA, US, etc.) in Figure 1 should be present in the footnote of Figure.

3.     The references should revise to the same format. Example Full page or Abbreviate page numbers. (Ref No 17 and 19)

Best wishes

Reviewer  
